# Evaluation of Enterococcal Probiotic Usage and Review of Potential Health Benefits, Safety, and Risk of Antibiotic-Resistant Strain Emergence

**DOI:** 10.3390/antibiotics12081327

**Published:** 2023-08-17

**Authors:** Eric Jeeho Im, Harry Hyun-Yup Lee, Minzae Kim, Myo-Kyoung Kim

**Affiliations:** 1College of Arts and Sciences, Washington University, St. Louis, MO 63130, USA; eric.im@wustl.edu; 2School of Osteopathic Medicine, Campbell University, Lillington, NC 27546, USA; 3College of Arts and Sciences, Boston University, Boston, MA 02215, USA; 4Thomas J. Long School of Pharmacy, University of the Pacific, Stockton, CA 95211, USA

**Keywords:** enterococci, probiotic, VRE, enterococcal consumption, virulence factor, antibiotic resistance

## Abstract

Enterococci are often used in probiotics but can also cause nosocomial infections. As such, enterococcal consumption may have beneficial health effects, but a thorough evaluation of virulence absence and risk of antibiotic resistance spread is needed at the strain level. This article reviewed ten online health product shopping websites in the US. On these websites, 23 probiotic products using enterococci were found across 12 companies. In addition, this article reviewed studies that demonstrated the probiotic potential of enterococcal consumption (e.g., gastrointestinal and respiratory disease, hyperlipidemia alleviation, as well as infection prevention). To investigate the safety aspects of enterococci, the present work examined studies evaluating virulence factors and antibiotic resistance. Furthermore, this article assessed research that explored these virulent factors, specifically in probiotics containing enterococci, as well as the potential transfer mechanism of their antibiotic resistance. Based on reviewed data, enterococcal probiotic consumption has been proven beneficial for conditions or symptoms of multiple diseases without any apparent adverse effects. However, due to the plasmid- or transposon-mediated gene transfer ability of enterococci, surveillance monitoring and further studies regarding enterococcal consumption are warranted. Future studies that identify enterococcal strains safe to use in probiotics without virulence factors and antibiotic resistance are imperative for evidence-based decisions by health organizations and government agencies.

## 1. Introduction

The genus *Enterococcus* consists of lactic acid bacteria (LAB) found predominantly in the gut of humans and animals. Among enterococcal species, *E. faecium* and *E. faecalis* are the predominant species of the human gastrointestinal system. Enterococci are also involved in the fermentation process of various foods, including cheeses and sausages [1]. Additionally, certain species of *Enterococcus* are utilized as probiotics to maintain healthy gastrointestinal microbiota and reduce gastrointestinal inflammation. They also demonstrate the ability to produce bacteriocins, which are proteins produced by bacteria to inhibit growth or kill other competing bacteria [2,3]. However, enterococcal strains can carry plasmid-mediated resistance genes, which can be transferred between bacterial species, causing decreased susceptibility to common antibiotics [4]. These plasmid-mediated genes in enterococci have contributed to Vancomycin-Resistant Enterococci (VRE), which are problematic in the clinical setting [5].

In the United States (US), the approval process for probiotics falls under the regulation of the US Food and Drug Administration (FDA) within the category of dietary supplements. Unlike pharmaceutical drugs, probiotics are dietary supplements and do not go through a pre-market approval process by the FDA. Instead, manufacturers bear the responsibility of ensuring the safety and accuracy of labeling for their probiotic products. Under the Dietary Supplement Health and Education Act (DSHEA) of 1994, manufacturers are obliged to comply with regulations such as Good Manufacturing Practices (GMPs) to guarantee the quality, purity, and safety of their products. While the FDA has the authority to act against unsafe or misbranded dietary supplements, they generally do not conduct pre-market testing or review of probiotics or other dietary supplements. Instead, the FDA can grant a designation called the Generally Recognized as Safe (GRAS) status. Substances that have previously received GRAS status since 1998 have been organized into a list made available for reference by industry members [6]. Companies seeking to use a specific probiotic strain can refer to the GRAS list to confirm whether the food additive has previously been recognized as safe for consumption by qualified experts [7].

A company seeking a GRAS status for an unreviewed strain must submit a notification to the FDA. The request must include comprehensive details about the substance and its intended use in food, along with scientific evidence supporting its safety. The GRAS panel then reviews the notification and supporting evidence to assess whether the substance meets GRAS criteria. Based on their evaluation, the FDA can either agree with the GRAS determination or express concerns and objections if safety issues arise or insufficient evidence is found. In case of objections raised by the FDA, the substance cannot be designated as GRAS. The company may be required to undertake further measures to prove safety or pursue formal approval as a food additive [7,8,9].

Regarding probiotics, GRAS status can only be attributed to a single strain for a specific application of a probiotic product, not for the whole species. For example, *Lactobacillus acidophilus* CBT LA1 and *Bacillus subtilis* ATCC SD-7280 have obtained GRAS status. However, the status does not apply to all *L. acidophilus* or *B. subtilis* strains.

No strains of enterococci have been granted GRAS [6]. As GRAS status is not required to market a product, probiotic products containing enterococci can still be on the market [8]. Therefore, there is a need to assess the potential efficacy and safety of enterococci as probiotics. This review will discuss the current prevalence of enterococci usage in probiotic products, efficacy, safety, virulence factors, antibiotic resistance, and potential transfer of antibiotic resistance among enterococcal strains.

## 2. Current Usage of Enterococci in the US and Regulations

Given that probiotics conserve or restore beneficial bacteria populations in the gut flora, an increasing effort has been made to utilize probiotics due to the health benefits that probiotic bacteria can provide. To date, the most well-established and conventionally utilized probiotic strains include lactobacilli and bifidobacteria [10,11]. However, it is important to explore the safety of additional bacterial strains that are also included in probiotic products to determine whether these strains could have unique added benefits that outweigh the potential harms. Among these microorganisms, enterococcal species remain prominent due to their ability to maintain healthy gut flora and reduce gastrointestinal diseases. Aside from the benefits of enterococci, they also demonstrate the ability to transfer both antibiotic resistance and virulent genetic material, warranting further investigation [5].

There is considerable commercial interest in probiotics since millions of people consume them daily [12]. To measure the extent of dietary consumption of enterococci, the present work reviewed ten widely used online dietary shopping sites in the US, and identified probiotics for humans that contained the enterococcal species *E. faecium* and *E. faecalis*. The following online shopping sites were selected: Amazon [13], GNC [14], Vitamin Shoppe [15], Bodybuilding.com [16], The Vitamin Company [17], iHerb [18], Swanson Vitamins [19], Lucky Vitamin [20], PureFormulas [21], and Thrive Market [22]. These websites were selected according to the following criteria: variety and breadth of products, reputation, affordability, reliability, and speed of shipping. Table 1 provides a summary of the enterococcal species and strains used in 23 products in the US in 2023.

To date, the safety of enterococcal strains in food and food supplements has not been proven by international or national government agencies despite recent advances in scientific knowledge of the bacteria. As indicated previously, the US FDA does not grant *Enterococcus* GRAS status [6]. Furthermore, *Enterococcus* is not included in the Qualified Presumption of Safety (QPS) list from the European Food Safety Authority (EFSA) [23].

Despite the wide use of enterococci in probiotics, the safety of enterococcal strains in food supplements should be evaluated at the strain level because studies have only shown that nosocomial enterococcal subtypes are genotypically different from harmless subtypes used in food. For instance, Montealegre et al. divided *E. faecium* into three subtypes: clade A1, found in clinical settings; clade A2, found in animals; and clade B, found in healthy individuals [24]. Similarly, Beukers et al. conducted a comparison of the complete genomes of *E. faecium.* Their findings indicated that commensal and clinical isolates exhibit distinctive clustering patterns, implying that these strains may have adapted to their particular environments [25]. The findings suggest a push for updated legislation regarding the probiotic use of *Enterococcus* to distinguish between pathogenic and commensal strains.

## 3. Probiotic Potential (Efficacy of Enterococcal Strains)

This section discusses the probiotic potential of enterococci. Table 2 summarizes clinical trials and animal studies that demonstrated the efficacy of enterococcal consumption.

### 3.1. Effects on Gastrointestinal (GI) Diseases

Although only a few studies have investigated the potential probiotic effects of enterococci on the GI system (Table 2), probiotic enterococcal strains demonstrated therapeutic effects on GI diseases. The beneficial effect of enterococci on treating diarrhea has been relatively well-documented. In a double-blind, randomized trial, Buydens and Debeuckelaere investigated the efficacy of *E. faecium* NCIMB 10415 (SF68^®^) in the treatment of 211 adults with acute diarrhea. Patients who were administered SF68 experienced a significantly shorter number of days with diarrhea compared to those who received a placebo. There were statistically significant differences between the two treatments (*p* < 0.01). By the third day of treatment, diarrhea was present in only 8% of SF68-treated patients compared to 66% of placebo-treated patients. The average duration of diarrhea was 1.69 days (with an SD of 0.6) in the SF68 group, whereas it was 2.81 days (with an SD of 0.9) in the placebo group. Pathogens identified in the initial stool culture (e.g., *Salmonella* spp., *Campylobacter* spp., and *Yersinia* sp.) were no longer detectable in the post-treatment examination. No adverse reactions were observed during the study [26]. Chen et al. also demonstrated the positive impact of *E. faecalis* (in BIO-THREE^®^ probiotic preparation, which also contains *Bacillus mesentericus* and *Clostridium butyricum)* on acute infectious diarrhea in 304 pediatric inpatients [27]. The average duration of diarrhea after initiating treatment was 60 h in the group receiving probiotics, whereas it was 86 h in the group receiving a placebo (*p* = 0.003). Thus, the probiotics group had a shorter hospital stay compared to the placebo group (*p* = 0.009). The probiotics group also showed an increase in IL-10 in both serum and cell culture supernatants. Since IL-10 is a regulatory cytokine that inhibits both antigen presentation and the release of proinflammatory cytokines, IL-10 is proposed to exert anti-inflammatory effects. Chen et al. also demonstrated a slightly lower level of tumor necrosis factor-alpha, Interferon-gamma, and IL-12 in the probiotics group in comparison to the placebo group. These cytokines generally promote inflammations and/or immune responses [27].

Irritable Bowel Syndrome (IBS) is also a common gastrointestinal disorder. Its symptoms include abdominal pain, flatulence, and irregular bowel movements. In an open-label trial with 85 IBS patients, Fan et al. found that a probiotic treatment containing enterococci, lactobacilli, and bifidobacteria improved stool characterization, urgency, distension, pain, duration, and frequency without any adverse drug reactions (ADRs). The improvement of symptoms persisted when outcomes were measured 2 weeks after stopping treatment. Interestingly, the probiotic treatment significantly decreased *Enterococcus* and *Bacteroides* counts (*p* < 0.05) in the intestinal flora, whereas the count of lactobacilli was significantly increased (*p* < 0.01) after the treatment. The authors were not able to clearly explain the reason for the reduction in enterococcal count reduction despite the consumption of enterococci through the probiotic capsules. They speculated that unknown intestinal flora homeostasis mechanisms might contribute to the reduction. Although the treatment did not significantly reduce *C. difficile* and *Enterobacteriaceae* counts (*p* > 0.05) due in part to the small sample size, the colony-forming units (CFU) of *C. difficile* and *Enterobacteriaceae* in the intestinal flora were reduced from 9.34 ± 0.91 to 8.97 ± 0.97 and 9.33 ± 0.81 to 9.30 ± 0.77, respectively. The study suggested that the improvement in symptoms may be due to a change in the intestinal microbiome [28]. In an open randomized placebo-controlled trial consisting of 62 patients with post-infectious irritable bowel syndrome (PI-IBS), Yakovenko et al. came to a similar favorable conclusion. Histological examination of the colon mucosa showed signs of a low degree of inflammation in all patients. However, when treated with Bifiform^®^, a probiotic containing *E. faecium* ENCfa68, a moderate increase in the level of fecal calprotectin was found in 62.2% of patients with colonic dysbiosis. Most patients in the treatment group showed favorable clinical outcomes such as restoration of the normal composition of intestinal microbiota and normalization of fecal calprotectin content at the end of course therapy and 6 months post-treatment [29].

Ahmadi et al. found that a probiotic cocktail containing *five* strains of lactobacilli and five strains of enterococci reduced inflammation in mice [30]. The treatment increased physical function and reduced the development of intestinal dysbiosis, leaky gut, inflammation, and metabolic dysfunctions in mice that were on a high-fat diet. The primary mechanism through which the probiotics exerted their beneficial effects was by modulating the gut microbiota. This led to an increase in the integrity of tight junctions, thereby reducing gut permeability and subsequently lowering inflammation. These findings indicated that probiotic treatments may have the potential to prevent or alleviate age-related intestinal permeability and inflammation in older individuals. Although this study investigated a mixture of various bacterial species, the results of the study support the potential of enterococci in the treatment of various gastrointestinal diseases, like diarrhea and IBS.

### 3.2. Effects on Respiratory Diseases

Probiotic enterococci may also reduce symptoms in children with allergic rhinitis (AR). In a randomized placebo-controlled trial with 250 children, a probiotic mixture containing *E. faecium* L3, *Bifidobacterium animalis,* and *Lactis* BB12 significantly (*p* < 0.01) reduced Nasal Symptoms Score (NSS) after treatment. The restoration of intestinal microbiome homeostasis had immunomodulatory and anti-inflammatory benefits, which may have been responsible for reducing the allergic reactions characterizing AR. Researchers demonstrated that the probiotic treatment significantly (*p* < 0.01) reduced side effects and the need for corticosteroids and antihistamines [31]. Di Pierro et al., in a retrospective study, also demonstrated a significant reduction (*p* < 0.001) of rhinitis, watery eyes, and cough-bronchospasm when a probiotic mixture of *E. faecium* L3, *B. animalis,* and *Lactis* BB12 was given to atopic children. They also demonstrated that the treatment significantly reduced the need to use oral antihistamines, as well as inhaled and systemic corticosteroids [34].

In addition, recurrent bronchitis can be improved by probiotic enterococci. In a double-blind placebo-controlled multicenter trial, Habermann et al. demonstrated the benefit of the cells and autolysate of *E. faecalis* strain (Symbioflor 1) of human origin in 136 patients with chronic recurrent bronchitis. Duration until relapse was significantly longer in the treatment group (699 days) than in the placebo group (334 days) (*p* = 0.01). The severity of relapses in the treatment group was also reduced significantly (*p* = 0.001). Only four patients in the treatment group required antibiotic therapy compared to 13 patients in the placebo group [32].

### 3.3. Antimicrobial Effects

Bacteriocins are proteinaceous compounds produced by bacteria that have antimicrobial activity against microorganisms of the same species or species related to the bacteriocin-producing strain [38]. Given bacteriocins’ low toxicity and bactericidal activity, as well as the rise of antibiotic-resistant bacteria, bacteriocins may be a novel potential solution for combating antibiotic-resistant infections [38]. A bacteriocinogenic gene cluster of *E. faecium* E86 encoding enterocin P (EntP) (bacteriocins produced by *Enterococcus*) was able to inhibit the growth of *Listeria monocytogenes* and VRE strains [39].

Farias et al. measured the effectiveness of EntP against 25 *L. monocytogenes* and 14 *E. faecium,* and *E. faecalis* isolates. These enterococci isolates are representative of clinical samples of cases of infection or colonization in humans. They found that all tested *L. monocytogenes* and enterococcal strains were sensitive to EntP activity. In addition, EntP demonstrated bacteriolytic activity against pathogenic enterococcal strains, causing continuous growth reduction [39]. Bacteriocin producers may also stimulate the growth of beneficial bacterial species in the gastrointestinal tract. Bhardwaj et al. demonstrated that the bacteriocinogenic strain, *E. faecium* KH24, increased levels of lactobacilli in mice feces, which enhanced intestinal barrier defense. Two groups of mice were given bacteriocin-producing *E. faecium* KH24 (Bac+) and a non-bacteriocinogenic variant (Bac-), respectively, for 12 days. In this period, the observed fecal counts of lactobacilli were significantly higher (*p* < 0.05) in the Bac+ group. Furthermore, *E. faecium* KH24 did not show any pathogenic characteristics nor transferable antibiotic resistance but demonstrated resistance to GI stress and an ability to sufficiently produce bacteriocins [2]. These findings suggested the potential antimicrobial effect of bacteriocinogenic enterococci.

Although the production of bacteriocins by enterococci in probiotics has not been investigated, the reduction of acute respiratory infections by enterococcal consumption was reported by an observational study. Gonchar et al. observed the incidence of acute respiratory infections in two groups in an orphanage: a group treated with daily *E.faecium* L3 suspension and a no-treatment group. The average number of acute respiratory infections (ARI) cases per child were 0.29 ± 0.13 in the group with enterococcal consumption and 0.73 ± 0.12 in the control group (*p* < 0.05) [35].

In addition to the potential production of bacteriocins, enterococcal consumption may stimulate the production of immune cells as well. In a *Salmonella*-infected piglet model, Rieger et al. demonstrated that the consumption of *E. faecium* NCIMB10415 increased intraepithelial lymphocyte (IEL) number, which may potentially allow early detection of pathogenic bacteria [33]. Although further investigation is necessary for a more definitive conclusion, enterococcal consumption may have beneficial effects for preventing infections.

### 3.4. Hypocholesterolemic Effects

Probiotic enterococci have also shown hypocholesterolemic effects. In a double-blinded randomized and placebo-controlled human volunteer study, Hlivak et al. provided probiotic strain *E. faecium* M74 to volunteers in the treatment group. They demonstrated a significant reduction in LDL levels compared to the placebo group (3.85 ± 0.27 vs. 3.09 ± 0.21 mmol/L, *p* < 0.001) at the end of the 56-week treatment. However, no significant changes in HDL and triglyceride levels were noted [36]. A similar finding was also reported by Agerbaek et al. in a randomized, double-blind, and placebo-controlled study. A milk product containing *E. faecium* and two strains of *Streptococcus termophilus* were given to 58 non-obese and healthy volunteers for 6 weeks. LDL was significantly decreased by 10% (equivalent to −0.42 mmol/L) in the treatment group, while HDL and triglyceride levels remained unchanged (*p* < 0.01) [37].

### 3.5. GABA-Production

Psychobiotics represent a new category of probiotics that enhance mental well-being by producing neuroactive substances like GABA. This molecule is a major inhibitory neurotransmitter in the brain. Reduced GABA levels are associated with neurological diseases like Alzheimer’s disease or neuropathic pain [40]. While the exact relationship between Alzheimer’s disease and GABA is not fully understood, a reduction of GABA levels in the hippocampus, alterations in the expression of GABA receptors, and a degeneration of GABAnergic interneurons were reported in patients with Alzheimer’s disease [38]. A particular GABA-producing enterococcal strain, isolated from the gut of marine shrimp, was tested in vitro for the production of GABA. Thin Layer Chromatography (TLC) analysis showed that the *Enterococcus* isolates exhibited high GABA production, suggesting its potential role in neurological modulation [41].

## 4. Safety of Enterococcal Consumption: Virulence and Antibiotic Resistance

### 4.1. Opportunistic Pathogenicity of Enterococci

Although enterococci are commensal organisms that are part of the natural human gut flora, they have emerged as common pathogens causing nosocomial infections, such as endocarditis, bacteremia, urinary tract infections (UTIs), intra-abdominal and pelvic infections. Nearly 80% of these infections were associated with *E. faecalis* [42]. Most enterococcal strains are harmless, but some of the strains found in clinical settings are pathogenic because hospitals serve as reservoirs for antibiotic-resistant strains [43].

Enterococci are opportunistic pathogens, meaning that they are not usually pathogenic, but they can cause infections in individuals with weakened immune systems. Enterococci are harmless in their natural habitats (GI tract) but can exhibit pathogenicity outside this anatomical site [44]. For instance, translocation across intestinal mucosal surfaces to other tissues and systems, like the lung, liver, spleen, lymph nodes, and circulatory system, has been linked to numerous diseases and disorders [5]. Manfredo-Vieira et al. demonstrated that the translocation of *E. gallinarum* into systemic organs induced autoimmune responses in patients with systemic lupus erythematosus and autoimmune hepatitis [45]. In addition, Wang et al. showed that the translocation of *E. faecalis* had potentially mutagenic and carcinogenic effects in a cell culture model through the production of clastogens (chromosome-breaking factors). For example, the clastogen superoxide (O_2_^−^) produced by *E. faecalis* mediated COX-2 expression, which caused macrophage-induced chromosomal instability and DNA damage in neighboring cells. [46].

### 4.2. Virulence Factors and Pathogenicity of Enterococcus in Probiotics

The existence of various virulence factors will lead to the pathogenicity of enterococci. These virulence factors can be classified by their functions: exoenzyme, adherence, exotoxin, immune modulation, and biofilm formation. Table 3 summarizes research investigating virulent factors and their functions.

Despite growing concerns over the pathogenicity of enterococci in hospital settings, no reports have demonstrated virulence factors in enterococcal probiotics. Domann et al. sequenced and compared *E. faecalis* contained in the probiotic Symbioflor 1^®^ to clinical VRE isolates. *E. faecalis* strains in Symbioflor 1 were found to lack gene coding for essential virulence factors, such as *cyl* and *esp.* The *E. faecium* T110, a component of the probiotic BIO-THREE^®^, was sequenced completely and compared to pathogenic and non-pathogenic enterococcal strains. The gene encoding virulence factors were not found in the genome of *E. faecium* T110. The genome was noticeably different from the pathogenic strains found in hospitals, which included VRE and pathogenicity genes. Of the 40 enterococcal virulence genes established in the VFDB database (http://www.mgc.ac.cn/VFs/main.htm (accessed on 30 May 2023), 32 genes were missing in the T110 genome, affirming its safety as a non-pathogenic strain. The eight virulence genes present in the genome were not well-characterized and did not seem to contribute much to its pathogenicity [62].

### 4.3. Antibiotic Resistance

Avoparcin, a glycopeptide antibiotic similar to vancomycin, has been used as a growth promoter in livestock feed since the 1970s. The use of avoparcin coincided with the emergence of VRE outbreaks in hospitals. This chain of events suggests the possibility of VRE originating from antibiotic use in livestock, which subsequently spread to humans and hospitals through consumption [63].

Enterococcal species can be divided into two major categories based on their susceptibility to antibiotics: clade A and clade B. Clade B strains are susceptible to antibiotics (i.e., ampicillin and vancomycin) and, therefore, are not problematic in treatment against enterococci. Clade A strains, on the other hand, are hospital-derived strains that have diverged from Clade B strains and have adapted antibiotic resistance to various antibiotics, such as ampicillin and vancomycin [63,64]. Persistent use of antibiotics in hospitals and veterinary medicine has created multi-drug-resistant strains of enterococci that may be problematic [5]. These multi-drug resistant strains may spread in hospital settings, mainly through the hands of healthcare workers as well as medical equipment, leading to problematic infections [43].

Of particular concern is the increase in the incidence of vancomycin-resistant strains, with over 30% of all nosocomial enterococcal infections having resistance to this antibiotic as of 2019 (CDC). Vancomycin is often used to treat ampicillin-resistant pathogens [42]. Therefore, VRE, when pathogenic, has been associated with higher mortality in patients after undergoing hematopoietic cell transplantation [65]. Vancomycin resistance is encoded by specific gene clusters with nine variants (*vanA*, *vanB*, *vanC*, *vanD*, *vanE*, *vanG*, *vanL*, *vanM*, *vanN*), which can be identified by PCR or DNA sequencing. [64]. The different variants display varying levels of resistance to vancomycin and similar glycopeptide antibiotics. *VanA* and *B* are commonly linked to high levels of resistance, whereas *vanC*, *D*, *E*, *G*, *L*, *M*, and *N* typically demonstrate resistance at lower levels. Each variant employs a unique mechanism to confer resistance to vancomycin. For instance, *vanA*, *B*, *D*, *E*, *L*, *M*, and *N* are associated with the production of modified peptidoglycan, resulting in reduced affinity towards vancomycin. On the other hand, *vanC* generates an altered target site that vancomycin struggles to bind effectively, while *vanG* combines the synthesis of modified peptidoglycan precursors with the alteration of the target site. Complete sequencing of two probiotic strains, *E. faecium* T110 (BIO-THREE^®^) and *E. faecalis* (Symbioflor 1), showed the absence of antibiotic resistance genes [62,63].

Linezolid, daptomycin, and tigecycline are medications that can treat VRE. However, resistance to these three agents has emerged. According to the global Zyvox Annual Appraisal of Potency and Spectrum (ZAAPS) linezolid surveillance monitoring program, the number of linezolid-resistant enterococci (LRE) isolates has increased from 420 in 2002 to 813 in 2014. Similarly, the US Linezolid Experience and Accurate Determination of Resistance (LEADER) surveillance monitoring program reported that LRE *faecium* increased from 428 in 2004 to 589 in 2014, while LRE *faecalis* isolates increased from 196 to 239 in 2014 [66]. The prevalence of *E. faecalis* strains resistant to linezolid was found to be 2.8% in Asia, whereas, in the Americas, the prevalence of linezolid-resistant *E. faecium* was observed to be 3.4% [67]. The primary causes of LRE involve changes in the genetic material of the bacteria. These changes occur through mutations in specific genes responsible for producing 23S ribosomal RNA and regulatory genes that encode ribosomal proteins, namely *rplC*, *D*, and *lV*. These mutations result in the replacement of certain amino acids in various ribosomal proteins, including L3, L4, and L22 [68,69,70].

Daptomycin-resistant *Enterococcus* (DRE) has also been reported. Based on a meta-analysis in 2021 by Dadash et al., the prevalence of DRE (9%) is higher than that of LRE (2.2%). Multiple DRE mechanisms of resistance have been reported. However, two major categories of genes are found in DRE. These consist of regulatory genes for cell-envelope homeostasis and stress response, as well as genes that code for enzymes involved in the phospholipids of the cell membrane. Genes found in DRE *faecalis* are *cls*, *liaFSR*, and *gdpD*, whereas genes found in DRE *faecium* are *cls*, *liaSR,* and *yycFG* [71].

Tigecycline has been marked as a potential treatment option for complex soft tissue and intra-abdominal infections. However, it cannot be used for bloodstream infections due to inadequate antibiotic concentration in the bloodstream [72]. Dadash et al. also reported tigecycline-resistant *Enterococcus* (TRE) prevalence rates in Europe. TRE *faecium* was 3.9% and TRE *faecalis* was 0.4 [67]. Major genes that cause TRE are *tet (M)* and *tet (L). Tet (M*) is a ribosomal protection protein that alters the binding site of tigecycline, whereas *tet (L)* is an MFS-type efflux pump [71,72]. These tetracycline resistance genes also confer resistance to tigecycline [73]. Regarding the antibiotic-resistant risk of probiotic enterococci, no published research that sequences probiotic enterococci for the detection of these LRE, DRE, or DRE-related genes has been conducted.

### 4.4. Concern of Transfer of Virulence and Antibiotic Resistance

Enterococcal strains currently used in probiotics are not pathogenic, nor confer resistance to antibiotics. However, there is growing concern regarding the potential transfer of virulence and antibiotic-resistance genes between different enterococcal strains. Enterococci have a notable characteristic of possessing mobile genetic elements such as plasmids and transposons, which facilitate the efficient transfer of genes. This feature drives the evolution of certain strains, enabling them to adapt to different antibiotics found in clinical settings [74]. For instance, enterococcal strains can transfer genetic material like antibiotic resistance or virulence factors to each other or to other strains through the transfer of conjugative plasmids [5].

An example of such a transfer occurred in a study by de Niederhäusern et al., where they successfully transferred the *vanA* gene (associated with vancomycin resistance) from VRE to *Staphylococcus aureus* through the horizontal transfer of the Tn1546 transposon containing *vanA*. This discovery raised concerns about the horizontal transfer of vancomycin resistance, highlighting that VRE strains are capable of transferring their resistance to other pathogenic strains [75]. Another instance involved a clade B-classified *E. faecium* strain without vancomycin resistance. This clade B strain was found to possess the *vanN* gene and exhibited inducible vancomycin resistance. This finding suggests the potential transfer of vancomycin resistance from clade A to clade B strains, indicating that the extent of vancomycin resistance may be underestimated, especially in enterococcal strains previously described as lacking antibiotic resistance [64].

The highly efficient mechanism of gene transfer implies that a harmless enterococcal strain can acquire virulence or antibiotic resistance through conjugation with a pathogenic strain. In the case of probiotics, where enterococci are consumed in significant quantities, a large population of recipient bacteria is available for the transfer of virulence or antibiotic-resistance genes. This transfer can occur, for example, from pathogenic strains present in the human gastrointestinal tract to harmless probiotic strains [76]. Such transfer events can lead to the evolution of pathogenic or antibiotic-resistant probiotic strains, which can potentially cause problematic infections.

Olanrewaju et al. showed that conjugal transfer of resistance genes could result in an effect of biofiltration in the guts of zooplankton *Daphnia magna* and *D. pulex*. PCR and DNA sequencing was used to confirm that filter feeding in aquatic environments could lead to in vivo conjugative transfer of *vanA* resistance genes in *Daphnia*. These results showed that host enterococcal strains in *Daphnia* can acquire *vanA* simply through the consumption of *vanA*-containing bacteria in the aquatic environment. Such conclusions raise the possibility of humans being the end host of resistant enterococcal strains through the food chain [77].

In addition, Moubareck et al. demonstrated in vitro and in vivo conjugative transfer of the *vanA* resistance gene from vancomycin-resistant enterococcal strains isolated from pigs to vancomycin-susceptible human fecal isolates in gnotobiotic mice. The transfer event occurred in human isolates only 5 h after inoculation with the donor strain, suggesting that human bacteria may be able to acquire vancomycin resistance from enterococci of animal origin in a short time frame [78].

These findings suggest that the transfer of vancomycin resistance to human hosts is possible. Therefore, it is crucial to carefully monitor enterococcal strains currently deemed “safe” for any potential emergence of virulence or antibiotic resistance. Genetic changes can render these strains pathogenic at any time point in the future [74].

## 5. Future Directions and Conclusions

Enterococci have demonstrated probiotic characteristics, but also harbor potential to be pathogenic and resistant to commonly used antibiotics. An increasing number of studies have reported the advantages that enterocin-producing enterococcal strains can have on human health. Additionally, many studies have demonstrated the association between enterococcal strains and their beneficial effects on cholesterol reduction, the immune system, the gastrointestinal system, and respiratory allergies. However, enterococci for use as probiotics harbor safety concerns regarding the pathogenic and antibiotic-resistant nature of some of their strains and the potential to transfer these genes to other strains. To optimally utilize enterococci as probiotics, research is warranted to differentiate pathogenic species and strains with virulence from beneficial strains with probiotic potential. In addition, more studies on virulence and antibiotic resistance transfer to probiotic strains are needed. These studies would support future guidelines regarding safety-proven enterococcal strains. Additionally, enterococcal strains currently considered safe should be monitored for the emergence of pathogenicity driven by the acquisition of virulence and antibiotic-resistance genes between strains. These measures can help industries to be more willing to utilize enterococci in their products, as well as curb the spread of pathogenic lineages. Thus, studies commented in this review indicated that the usage of enterococci as probiotics still requires extensive evaluation of the safety status of the strains used.

## Figures and Tables

**Table 1 antibiotics-12-01327-t001:** Probiotics brands and products available for sale in 10 major US online dietary supplement shopping sites that contain *E. faecium* and *E. faecalis*.

Species and Strain	Company	Product	Online Store	Other Species Present
*E. faecalis* TH10	Dr. Ohira, Premier Research Labs, Quantum Nutrition Labs	Dr. Ohira’s Probiotics Professional Formula, Dr. Ohira’s Probiotics Original Formula, Premier Probiotic Caps, Quantum Probiotic Support	iHerb, Amazon, Swanson Vitamins, Lucky Vitamins, Pure Formulas	*Bifidobacterium breve* M16, *Lactobacillus acidophilus* ATCC SD521
*E. faecium* R0026	Natural Factors	ReliefBiotic IB, IBS Relief Biotic	iHerb, Amazon	*Lacticaseibacillus rhamnosus* R0011, *Lactobacillus helveticus* R0052, *Bacillus subtilis* R0179
*E. faecium* SD5843	ProBioCare	Probiotic for Men, Probiotic for Women	Vitamin Shoppe	*Lacticaseibacillus casei* LC11, *Lactiplantibacillus plantarum* LP115, *L. rhamnosus* GG
*E. faecium* T110	Advanced Orthomolecular Research	Probiotic 3	Amazon, iHerb, Pure Formulas	*Clostridium butyricum* TOA, *B. subtilis* TOA
*E. faecium* VPro21	Solaray	Mycrobiome Probiotic Adult 50+, Mycrobiome Probiotic Men’s, Mycrobiome Probiotic Women’s, Mycrobiome Probiotic Urgent care, Mycrobiome Probiotic Weight, Mycrobiome Probiotic Colon, Super Multidophilus	Vitamin Shoppe, Amazon, Pure Formulas, iHerb, Swanson Vitamins	*Bifidobacterium lactis* VK2, *B. infantis* VPro53, *B. longum* VPro51, *B. longum* VPro55, *B. breve* VPro52, *L. plantarum* VPro10, *Lacticaseibacillus paracasei* VPro224, *L. paracasei* VK4, *Lactococcus lactis* VPro17, *Lactobacillus gasseri* Vpro16
*E. faecium* W54	North American Herb and Spice, Zenement	Health-Bac, Proactiflora	Amazon	*B. lactis* W51, *B. lactis* W52, *B. longum* BL21, *B. lactis* BLa80 *L. acidophilus* W22, *L. paracasei* 20, *L. acidophilus* LA85, *L. plantarum* Lp90
*E. faecium* NS *	NatureWise	Time Release Probiotics, Maximum Care	iHerb	*L. casei*, *Limosilactobacillus fermentum*, *L. plantarum*
*E. faecium* NS *	Professional Formulas	IntestiCalm	Amazon, Pure Formulas	*L. rhamnosus*, *L. plantarum*, *Bifidobacterium bifidum*, *B. infantis*
*E. faecium* NS *	Nutra Biogenesis	MicroBiotic Intensive, MicroBiotic Lower GI	Amazon, Pure Formulas	*L. plantarum*, *L. paracasei* *B. lactis*, *B. longum*

* NS = Not Specified.

**Table 2 antibiotics-12-01327-t002:** Research that demonstrated the probiotic potential of enterococcal strains and their key functions.

Probiotic Potential	Type of Research	Enterococcal Strains	Functions	Reference
Acute Diarrhea	A placebo-controlled trial in adults	*E. faecium* SF68	Significantly shorter duration of acute diarrhea with no adverse drug reactions	Buydens et al. 1996 [26]
	A placebo-controlled trial in pediatrics	*E. faecalis* (in BIO-THREE^®^)	Significantly shorter duration of acute diarrhea and hospital stay Decreased levels of cytokines IL-10, TNF-α, IFN-γ, and IL- 12	Chen et al. 2010 [27]
Irritable Bowel Syndrome (IBS)	Open-label trial in adults	*Enterococcus* (non-specified strain)	Improvement of IBS symptoms Reduction of enterococcal count (*p* < 0.01) and count of *Bacteroides* (*p* < 0.05) in the intestinal flora	Fan et al. 2006 [28]
	Open-randomized placebo-controlled trial	*E. faecium* ENCFa68	Improved clinical manifestations of the disease Restoration of the normal composition of intestinal microbiota and normalization of the content of fecal calprotectin	Yakovenko et al. 2022 [29]
Gastrointestinal Inflammation	Mice	5 *Enterococcus* strains (not specified)	Reduced intestinal epithelial permeability by increasing stimulation of tight junctions Reduced inflammation	Ahmadi et al. 2020 [30]
Allergic Rhinitis (AR)	A placebo-controlled trial in pediatrics	*E. faecium* L3 LMG P27496	Reduced symptoms of AR Significant reduction in nasal symptom score Significant reduction in intake of pharmacological therapy (antihistamines and local steroids)	Anania et al. 2021 [31]
Chronic Recurrent Bronchitis	Double-blind, placebo-controlled multicenter trial	*E. faecalis* (in Symbioflor 1)	Significant reduction of the duration until relapse Significant reduction in the need for antibiotics	Habermann et al. 2001 [32]
Salmonellae-infections	Piglets	*E. faecium* NCIMB 10415	Increased number of intraepithelial lymphocytes (IEL), which are potentially related to the early detection of pathogenic bacteria	Rieger et al. 2015 [33]
Atopic respiratory symptoms	Retrospective trial in pediatrics	*E. faecium* L3	Significant reduction of rhinitis, watery eyes, and cough/bronchospasm Significant reduction of need for drugs (e.g., antihistamines, corticosteroids)	Di Pierro et al. 2018 [34]
Acute Respiratory Infections (ARI)	Observational research on orphan infants	*E. faecium* L3	Reduction of ARI cases	Gonchar et al. 2015 [35]
Hyperlipidemia	Randomized placebo-controlled human volunteers	*E. faecium* M74	Reduction in LDL cholesterol	Hlivak et al. 2005 [36]
		*E. faecium* (non-specified strain)		Agerbaek et al. 1995 [37]

**Table 3 antibiotics-12-01327-t003:** Virulence factors of enterococci classified by function.

Virulence Factors	Classification	Type of Research	*Enterococcus* strains	Functions	Reference
Adhesin to collagen of *E. faecalis* (Ace)	Adherence	Cell culture and analysis of clinical isolates from patients with endocarditis	*E. faecium* OG1RF	Ability to bind with collagen type I and IV, as well as laminin	Nallapareddy et al., 2000 [47]
Cell wall-anchored collagen membrane adhesin (Acm)	Adherence	Cell culture and analysis of isolates from patients with severe clinical infections	*E. faecium* TX0054 *E. faecium* TX2535 *E. faecium* TX2555	Ability to bind with collagen type I	Nallapareddy et al., 2003 [48]
Endocarditis- and biofilm-associated pili (Ebp)	Adherence	Analysis of isolates from rats	*E. faecalis* OG1RF	Contribution to biofilm formation and adherence to fibrinogen	Nallapareddy et al., 2006 [49]
Enterococcus collagen-binding adhesin (EcbA)	Adherence	Cell culture and analysis of hospital-acquired isolates in vitro	*E. faecalis* E1162	Ability to bind with collagen type I, IV, V, and fibrinogen	Hendrickx et al., 2009 [50]
*E. faecalis* antigen A (EfaA)	Adherence	In vitro analysis of clinical isolates from patients with endocarditis	*E. faecalis* EBH1	Potential function as an adhesin in the endocardium	Lowe et al., 1995 [51]
Extracellular surface protein (Esp)	Adherence	In vivo experiment regarding urinary tract infection with a mouse model	*E. faecalis* MMH94	Colonization and survival in the bladder	Shankar et al., 2001 [52]
Promotion aggregation complex (PrgB)	Adherence	In vitro experiment with PrgB (AS 10) wildtype and PrgB mutant	*E. faecalis* OG1RF	Promotion of aggregation and biofilm formation	Schmitt et al., 2018 [53]
Second collagen adhesin of *E. faecium* (Scm)	Adherence	Cell culture and analysis of endocarditis isolates	*E. faecium* TX0068 *E. faecium* TX0074	Ability to bind with specificity to collagen type V	Sillanpää et al., 2008 [54]
Serine glutamate repeat A (SgrA)	Adherence	Cell culture and analysis of hospital-acquired isolates in vitro	*E. faecium* U0317 *E. faecium* E1162	Potential contribution to biofilm formation by binding nidogen and fibrinogen	Hendrickx et al., 2009 [50]
Gelatinase (GelE)	Exoenzyme	Cell culture and analysis in vitro	*E. faecalis* (non-specified)	Cleavage of complement C3, resulting in activation of the complement system	Park et al., 2008 [55]
Serine Protease (SprE)	Exoenzyme	In vivo experiment with rabbit model of endophthalmitis	*E. faecalis* OG1RF	Activation of fsrABC by working together with GelE (Attenuation of endophthalmitis pathogenesis in rabbits with SprE-deficient mutant)	Engelbert et al., 2004 [56]
Cytolysin (Cyl)	Exotoxin	Translocation experiment in vitro	*E. faecalis* JH22, *E. faecalis* TX1322	Cleavage of complements C3 and iC3b	Zeng et al., 2005 [57]
		In vitro analysis of clinical isolates in patients	*E. faecalis* (non-specified)	Higher occurrence of Cyl among clinical pathogens	Huycke et al., 1995 [58]
Biofilm on plastic D (BopD)	Biofilm	In vivo experiment with mice	*E. faecalis* T9, *E. faecalis* 10D5, *E. faecalis* TDM	Contribution to biofilm formation	Creti et al., 2006 [59]
Quorum-sensing complex (FsrABC)	Biofilm	Comparative Transcriptional Analysis	*E. faecalis* OG1RF, *E. faecalis* TX5266	Encoding of a two-component signal transduction system for initiation of quorum sensing	Bourgogne et al., 2006 [60]
Capsular polysaccharides (Cps)	Immune Modulation	Cell culture and analysis in vitro	*E. faecalis* V583, *E. faecalis* LT02, *E. faecalis* LT06	Higher resistance to opsonophagocytosis	Thurlow et al., 2009 [61]

None of the above virulence factors have been reported in enterococci used in probiotics.

## Data Availability

It is a review article. All data were sourced from previously published articles or online websites listed in the references.

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
