# Peer review of "Evaluation of Enterococcal Probiotic Usage and Review of Potential Health Benefits, Safety, and Risk of Antibiotic-Resistant Strain Emergence"

_antibiotics, 2023, doi:10.3390/antibiotics12081327_

Round 1
Reviewer 1 Report
The manuscript describes the health benefit or risk of enterococcal probiotic usage comprising a detailed and systematic analysis of advantages and disadvantage of commercial products containing different strains of Enterococcus. The manuscript is a well-written critical appraisal of previous studies and literature in this and related fields of research.
The exposition is clear and detailed, is well designed and include useful discussions which provide important information for other researchers in the field. A range of methodological techniques have been used that demonstrates the presence antibiotic resistant genes in some isolates.
Overall, the level of literary presentation is of a high standard, the bibliography is comprehensive and correlates well with the main body of the text
In my opinion, the work presented meets the requirements of the publication by being discussed and put into context in relation to previous studies. The work has made a substantial and original contribution to knowledge highlighting the need to introduce resistance to antibiotics as a marker for safety products is of great interest as it threatens global health.
Author Response
I would like to express my sincere gratitude for your valuable time and thoughtful review of our manuscript. Thank you so much for your positve feedback.
Reviewer 2 Report
Reviewer assessment of Antibiotics 2455449
The review article entitled “The Enterococcus Probiotic Dilemma: Health Benefit or Resistant Risk? [Evaluation of Enterococcal Probiotic Usage and Review of Potential Health Benefits, Safety, and Resistant Risk]” deals with an interesting theme but it is too synthetic and vague in all parts to be informative. Therefore, all the aspects should be presented with more details. Coherence should be improved throughout the manuscript and more recent references should be consulted”. In addition, since the paper was submitted to “Antibiotics” the part on antibiotic resistance should be further extended
Specific remarks:
Line 14: an article cannot surveil anything. Moreover, it is not clear what “surveilled” means here. You rather listed the commercial products currently marketed. You should also mention which authorities approved the probiotic supplements.
Line 45: as probiotics
Line 62: they can be VRE, not be associated with VRE
Table 1: place where it is introduced. All table titles must be extended to better explain what the table presents.
Lines 109 and 113: which pathogens? Did the authors differentiate among the effect on diarrhea caused by different pathogens? If yes, summarize the observations
Lines 130-131: they administered enterococci and these decreased in gut?
Line 214: here and elsewhere the genus is spelled out only at first occurrence in the text, regardless of the species. It means, if I have “Enterococcus faecalis and Enterococcus faecium” I have to write “Enterococcus faecalis and E. faecium”
Line 227: a fermented milk… It is obvious that fermentation is biological
Line 235: how is GABA associated with Alzheimer disease?
Table 2: please, substitute arrows with text
Lines 249-250: is this the correct definition of “opportunistic pathogen”? Please, correct
Line 258: what are these clastogens? The agent names must be reported
Line 263: correct grammar
Line 266-268: this is a repetition
Section “virulence factors”: titles of subsections do not follow the journal style and should be modified. Moreover, the list of virulence factors is not complete and those listed are described too synthetically. Please, add the others with recent references.
Line 319: what is the meaning of “Other Effects: Cholesterol and GABA”? Why together? Cholesterol decreases while GABA is produced and is beneficial.
Lines 320-321: grammar
Table 3: functions of the virulence factor without the verbs, these should not be regular sentences (for the respective column the title “finding” is not meaningful). What is bacteremiai?
Lines 355-357: gene names in lower case (also lines 387-389). What are the variations? Variants maybe? What are the functions of the listed genes. i.e. the mechanism of resistance? These genes can be also detected by PCR according to many articles. Which are the genetic elements including van genes?
Lines 415-421: this is not in the correct place, maybe it should stay better at the beginning of the vancomycin resistance section as one of the causes of VRE emergence. It is not “an example from the 70s”, it is a prejudicial practice in animal rearing initiated in the 70s.
Line 432: please, enlarge this section because there are studies that demonstrated the transfer events in food and in vivo that should be added and commented.
Conclusions: as a perspective more studies on virulence and AR transfer on probiotic strains should be carried out
Grammar should be revised by a proficient English proofreader to correct many small mistakes
Author Response
I would like to express my sincere gratitude for your valuable time and thoughtful review of our manuscript. We appreciate your constructive feedback and the opportunity to address the concerns raised during the peer-review process.
We have carefully considered your comments and suggestions, which have enhanced the quality and clarity of our work. In response, we have made significant revisions to the manuscript, which are detailed below.
Additionally, the revised manuscript has been edited by a professional editor in the field of microbiology, as well as the director of the writing center at our institution. Spelling and grammar have been corrected throughout the manuscript.
Point 1: Line 14: an article cannot surveil anything. Moreover, it is not clear what “surveilled” means here. You rather listed the commercial products currently marketed. You should also mention which authorities approved the probiotic supplements.
Response 1: The word, “surveilled” was deleted and the sentence was modified as following in revised line 15: “This article reviewed ten online health product shopping websites in the U.S.”
In addition, the sentences below (revised lines 45-58) were added to the manuscript to describe the authorities that approved the probiotic supplements. We also added an explanation about the regulations of the probiotics:
In the United States, the approval process for probiotics falls under the regulation of the U.S. Food and Drug Administration (FDA) within the category of dietary supplements. Unlike pharmaceutical drugs, probiotics are dietary supplements and do not go through a pre-market approval process by the FDA. Instead, manufacturers bear the responsibility of ensuring the safety and accuracy of labeling for their probiotic products. Under the Dietary Supplement Health and Education Act (DSHEA) of 1994, manufacturers are obliged to comply with regulations such as Good Manufacturing Practices (GMPs) to guarantee the quality, purity, and safety of their products. While the FDA has the authority to act against unsafe or misbranded dietary supplements, they generally do not conduct pre-market testing or review of probiotics or other dietary supplements. The Generally Recognized as Safe (GRAS) status is a designation granted by the FDA to substances considered safe for consumption in food or as food additives. Regarding probiotics, certain strains of bacteria (e.g., Lactobacillus fermentum, Bacillus cereus) have obtained GRAS status, which has not been the case for Enterococcus strains [6,7].
Point 2: Line 45: as probiotics
Response 2: Modification to probiotics (plural form) has been made (revised line 56).
Point 3: Line 62: they can be VRE, not be associated with VRE
Response 3: “Their association with” was removed. The word, “VRE,” remains in revised line 74.
Point 4: Table 1: place where it is introduced. All table titles must be extended to better explain what the table presents.
Response 4: Table 1, 2, and 3 were moved where they were first introduced respectively. All table titles were revised to present more detail.
Point 5: Lines 109 and 113: which pathogens? Did the authors differentiate among the effect on diarrhea caused by different pathogens? If yes, summarize the observations
Response 5: The names of the pathogens (e.g., Salmonella sp., Campylobacter sp., and Yersinia sp.) have been clarified in revised line 126. Unfortunately, the authors did not differentiate the effect on diarrhea caused by different pathogens
Point 6: Lines 130-131: they administered enterococci and these decreased in gut?
Response 6: Yes. The following sentences are added to clarify in revised lines 145-151: “The improvement of symptoms persisted when measured 2 weeks after stopping treatment. Interestingly, the probiotic treatment significantly decreased Enterococcus and Bacteroides counts (p <0.05) in the intestinal flora whereas Lactobacillus count was significantly increased (p <0.01) after the treatment. The authors were not able to clearly explain the reason of Enterococcus count reduction despite consumption of enterococci through the probiotic capsules. They speculated that unknown intestinal flora homeostasis mechanisms may contribute to the reduction”
Point 7: Line 214: here and elsewhere the genus is spelled out only at first occurrence in the text, regardless of the species. It means, if I have “Enterococcus faecalis and Enterococcus faecium” I have to write “Enterococcus faecalis and E. faecium”
Response 7: The recommended modifications were made throughout the manuscript. (Table 2 and revised lines: 152, 153, 179, 186, 192)
Point 8: Line 227: a fermented milk… It is obvious that fermentation is biological
Response 8: The word, “biological,” is removed. (revised line 244)
Point 9: Line 235: how is GABA associated with Alzheimer disease?
Response 9: An explanation was added in revised lines 252-256: “While the exact relationship between Alzheimer’s disease and GABA is not fully understood, a reduction of GABA levels in the hippocampus, alterations in the expression of GABA receptors, and a degeneration of GABAnergic interneurons were reported in patients with Alzheimer’s disease [29]”
Point 10: Table 2: please, substitute arrows with text
Response 10: Arrows are substituted with words.
Point 11: Lines 249-250: is this the correct definition of “opportunistic pathogen”? Please, correct
Response 11: The definition is corrected in revised lines 268-269: “they are not usually pathogenic, but they can cause infections in individuals with weakened immune systems.”
Point 12: Line 258: what are these clastogens? The agent names must be reported
Response 12: The clastogen was clarified in revised lines 278-280: “the clastogen superoxide (⋅O2--) produced by E. faecalis mediated COX-2 expression, which caused macrophage-induced chromosomal instability and DNA damage in neighboring cells.”
Point 13: Line 263: correct grammar
Response 13: The grammar of the sentence has been corrected throughout the revised manuscript. In order to address Point 1, Line 263 in the previous version has been moved to revised lines 55-58. This was done to combine/condense repetitive explanations in a clear and succinct manner.
Point 14: Line 266-268: this is a repetition
Response 14: The repetitive lines are deleted.
Point 15:
Section “virulence factors”: titles of subsections do not follow the journal style and should be modified. Moreover, the list of virulence factors is not complete and those listed are described too synthetically. Please, add the others with recent references.
Response 15: The titles of the subsections have been corrected based on the journal style. Eleven additional virulence factors and related references were also added in Table 3. The classifications of virulence factors were also added. Some contents that were previously in section 4.3 have been moved/summarized in revised Table 3, making sure to delete repetetive sections. Additionally, former sections 4.2 and 4.3 were combined to streamline the article based on modifications. Please see Table 3 and lines 282-302 in the revised manuscript.
Point 16: Line 319: what is the meaning of “Other Effects: Cholesterol and GABA”? Why together? Cholesterol decreases while GABA is produced and is beneficial.
Response 16: There is not a specific relationship between cholesterol and GABA. In the previous manuscript, this section was meant to be an overarching “others” or “miscellaneous” summary. Nevertheless, in the revised manuscript, two separate sections have been made as there are no known relationships between these two effects : 3.4. Hypocholesterolemic effects and 3.5. GABA-producing effects (lines 236-259)
Point 17: Lines 320-321: grammar
Response 17: The sentence is corrected in revised lines 290-291.
Point 18: Table 3: functions of the virulence factor without the verbs, these should not be regular sentences (for the respective column the title “finding” is not meaningful). What is bacteremiai?
Response 18: The previous title, “Findings,” was modified to “Functions” in the revised new manuscript Table 3. The brief functions are summarized in noun forms, without the verbs. The word, “bacteremia,” was corrected to “urinary tract infection.”
Point 19: Lines 355-357: gene names in lower case (also lines 387-389). What are the variations? Variants maybe? What are the functions of the listed genes. i.e. the mechanism of resistance? These genes can be also detected by PCR according to many articles. Which are the genetic elements including van genes?
Response 19: Gene names are modifed to be written in lower case (revised lines 323-334 and 363-365). Further modifications were made to incorporate the suggestions. Please see revised lines 323-334 for the modifications.
Point 20: Lines 415-421: this is not in the correct place, maybe it should stay better at the beginning of the vancomycin resistance section as one of the causes of VRE emergence. It is not “an example from the 70s”, it is a prejudicial practice in animal rearing initiated in the 70s.
Response 20: The sentences are modified to incorporate the suggestions and moved to the beginning of the vancomycin resistance section in revised lines 304-308.
Point 21: Line 432: please, enlarge this section because there are studies that demonstrated the transfer events in food and in vivo that should be added and commented.
Response 21: Two additional studies are summarized in revised lines 397-410:
Olanrewaju et al. showed that conjugal transfer of resistance genes could result as an effect of biofiltration in the guts of zooplankton Daphnia magna and D. pulex. PCR and DNA sequencing was used to confirm that filter feeding in aquatic environments could lead to in vivo conjugative transfer of vanA resistance genes in Daphnia. These results showed that host enterococcal strains in Daphnia can acquire vanA simply through the consumption of vanA-containing bacteria in the aquatic environment. Such conclusions raise the possibility of humans being the end recipient of resistant enterococcal strains through the food chain [64].
In addition, Moubareck et al. demonstrated in vitro and in vivo conjugative trans-fer of vanA resistance gene from vancomycin resistant enterococcal strains isolated from pigs to vancomycin susceptible human fecal isolates in gnotobiotic mice [53b]. The transfer event occurred in human isolates only 5 hours after inoculation with the donor strain, suggesting that human bacteria may be able to acquire vancomycin re-sistance from enterococci of animal origin in a short time frame [65].
Point 22: Conclusions: as a perspective more studies on virulence and AR transfer on probiotic strains should be carried out
Response 22: Additional sentences (revised lines 429-434) were added to incorporate the suggestions: “In addition, more prospective studies on virulence and antibiotic resistance transfer to probiotic strains are needed.”
Once again, thank you very much for your thoughtful reviews. I hope our revisions address all your suggestions this time.
Reviewer 3 Report
Although the work is interesting and necessary for the wider use of enterococci as probiotic bacteria, it is necessary to be very careful about the correct spelling of the name of the microorganism. Furthermore, the references need to be adjusted according to the journal's instructions.

Author Response
I would like to express my sincere gratitude for your valuable time and thoughtful review of our manuscript. Thank you so much for your positive feedback. We carefully reviewed the manuscript to correct the spelling of the names of the microorganisms. In addition, the references have been adjusted according to the journal’s instructions. The revised manuscript has been edited by a professional editor in the field of microbiology, as well as the director of the writing center at our institution. Spelling and grammar have been corrected throughout the manuscript.
Round 2
Reviewer 2 Report
The review article entitled "The Enterococcus Probiotic Dilemma: Health Benefit or Re-sistant Risk? Evaluation of Enterococcal Probiotic Usage and Review of Potential Health Benefits, Safety, and Resistant Risk" was modified according to my observations and I found it improved but some issue must still be fixed. The authors state that grammar was thoroughly revised but there still are parts syntactically and grammatically incorrect, e.g. lines 68-70: "With the aforementioned beneficial effects of probiotics on health and eliminating pathogens that compete with gut flora, it is important to assess different candidate bacteria for probiotics.", what does this sentence mean? Please, note that who has to make English style corrections must be a mother tongue proofreader or editing service with scientific competence.
Parts to be improved:
Lines 55-58; The GRAS status is attributed to single strains for specific applications and not to the whole species. A proof of this is the B. cereus that you cite as an example: many strains of this species are pathogenic but some strains are safe and earned the GRAS status. Threfore, I would be very grateful to you if you can clarify very well this point, especially because in many scientific articles the GRAS status is reported improperly and it appears that some bacterial species are declared GRAS as a whole, but it is never so. When is the GRAS status attributed? Upon producer request for particular applications? If yes, this explains why no Enterococcus strains are GRAS yet.
Line 66: "to utilize probiotics due to its health benefits", what is "its" referred to?
Line 68: please, use the common term lactobacilli instead of Lactobacillus here and in the whole manuscript, since the taxonomy of the Lactobacillus genus was changed in 2020
Line 74: please, delete "as well as VRE"
Line 76: what do you mean with "to measure the scope". Maybe to measure the extent?
Lines 78-81: the url and the date of last access must be reported for each
Lines 86-87: why all in bold? Why capitalized initials (valid also for the title of Table 2 and section titles)? US without dots
Table 1: Please, delete (B.= Bifidobacterium, L= Lactobacillus, also because you have Bacillus subtilis with the "B"). Why "major species"? Better "other species present". For lactobacilli you have to use the new nomenclature and spell out each different genus at first occurrence, e.g. Lacticaseibacillus rhamnosus, Lactiplantibacillus plantarum. Last row: "L. paracasei"
LInes 90-94: it is not correct to say that "Enterococcus is not recommended", in fact it is included in food supplements, it is rather that safety of strains used in food and food supplements must be proven at the strain level by excluding virulence and antibiotic resistance traits. Please, report correctly these aspects
Line 104: stipulated? With whom? Rather: has set the guidelines to...
Line 107: genus in italics
Table 2: the species must not be capitalized. What is ADR? Last column: the references go with first author et al., also in Table 3
Line 113: why capital letters for gastrointestinal diseases?
Line 126: species plural is abbreviated spp. not sp., for Yersinia it can be only Yersinia enterocolitica I think
Line 128: reference number
Line 151: might. Please, note that when you refer to previous results or activities of researchers the correct form of verbs is the past tense.
Line 167: delete "microbiota". It must be intestinal dysbiosis
Line 192: of the E. faecalis strain of human origin...
Line 222: "the production of bacteriocins by probiotic enterococcal consumption", it is not the consumption that produces bacteriocins, Please, adjust
Line 226: what is E. faecium liquid? E. faecium suspension, maybe? What is ARI?
Lines 230-231: italicize Salmonella and E. faecium
Line 248: GABA production
Line 261: of enterococci, otherwise capitalize and italicize
Lines 363-366: please, add a sentence to explain that these tetracycline resistance genes also confer resistance to tigcycline
Line 371: antibiotic resistance genes
Line 421: their beneficial effects on cholesterol reduction...
Line 426: delete "beneficial species", you are referring to strains, not species
Lines 430-431: driven by the acquisition of virulence and antibiotic resistance genes.
As expressed in comments and suggestions
Author Response
Please see the attached document.
#####################################################
I would like to express my sincerest gratitude for your valuable time and additional review of our manuscript. We appreciate your constructive feedback.
We have carefully considered your comments and suggestions. In response, we have made significant revisions to the manuscript, which are detailed below.
Regarding English, two of the authors’ primary language is English. Additionally, the professional editor and the director of the writing center for the previously revised manuscript are all only English speakers. However, after reviewing your suggestions and examples, we agree there are still areas to further improve. We also carefully reviewed the two websites (https://wwwnc.cdc.gov/eid/page/scientific-nomenclature and https://wwwnc.cdc.gov/eid/page/scientific-nomenclature) and further revised the manuscript. In addition, we also incorporated another editor’s suggestions.
We will also attach a file that shows all the “track changes” on (both in a word file and pdf file) in addition to the clean file for the second revision.
Point 1: Lines 55-58; The GRAS status is attributed to single strains for specific applications and not to the whole species. A proof of this is the B. cereus that you cite as an example: many strains of this species are pathogenic but some strains are safe and earned the GRAS status. Therefore, I would be very grateful to you if you can clarify very well this point, especially because in many scientific articles the GRAS status is reported improperly and it appears that some bacterial species are declared GRAS as a whole, but it is never so. When is the GRAS status attributed? Upon producer request for particular applications? If yes, this explains why no Enterococcus strains are GRAS yet.
Response 1: We made major revisions to incorporate your suggestions as shown in lines 55-79. When we double-checked the GRAS list, no strain of B. cereus was listed in the GRAS notice website. To ensure accuracy, Bacillus subtilis ATCC SD-7280 was used as an example instead of B. cereus.
Point 2: Line 66: "to utilize probiotics due to its health benefits", what is "its" referred to?
Response 2: The sentence has been modified to “the health benefits that probiotic bacteria can provide” (new lines 82-83).
Point 3: Line 68: please, use the common term lactobacilli instead of Lactobacillus here and in the whole manuscript, since the taxonomy of the Lactobacillus genus was changed in 2020
Response 3: Lactobacillus has been modified to lactobacilli throughout the manuscript. For consistency, Bifidobacterium has also been modified to bifidobacteria (new line 84). Other parts of the manuscript have been modified accordingly. Please see the attached file with “track changes” on for details.
Point 4: Line 74: please, delete "as well as VRE"
Response 4: The phrase has been removed.
Point 5: Line 76: what do you mean with "to measure the scope". Maybe to measure the extent?
Response 5: The paragraph has been modified to “To measure the extent of dietary consumption of enterococci.”
Point 6: Lines 78-81: the url and the date of last access must be reported for each
Response 6: The references (including the urls and access dates) have been added.
Point 7: Lines 86-87: why all in bold? Why capitalized initials (valid also for the title of Table 2 and section titles)? US without dots
Response 7: The bolded words have been changed to regular font. We also modified the capitalized initials to regular lower-case fonts. We also removed the dots from US throughout the manuscript. All table titles (Table 1, 2, and 3) have been modified.
Point 8: Table 1: Please, delete (B.= Bifidobacterium, L= Lactobacillus, also because you have Bacillus subtilis with the "B"). Why "major species"? Better "other species present". For lactobacilli you have to use the new nomenclature and spell out each different genus at first occurrence, e.g. Lacticaseibacillus rhamnosus, Lactiplantibacillus plantarum. Last row: "L. paracasei"
Response 8: A major revision was made in the last column of table 1. All organism names are double-checked based on the ISAPP website (https://isappscience.org/) and PubMed.
Point 9: Lines 90-94: it is not correct to say that "Enterococcus is not recommended", in fact it is included in food supplements, it is rather that safety of strains used in food and food supplements must be proven at the strain level by excluding virulence and antibiotic resistance traits. Please, report correctly these aspects
Response 9: The modification was made as recommended (in lines 107-109 in the revised manuscript).
Point 10: Line 104: stipulated? With whom? Rather: has set the guidelines to...
Response 10: Lines 104-106 in the previous manuscript have been retracted/deleted. After double-checking the reference, we realized that the sentence in the previous manuscript was not accurate.
Point 11: Line 107: genus in italics
Response 11: Instead of using the genus, we modified to “enterococcal strains” to keep consistency in the phrasing within the manuscript. (in line 128 in the revised manuscript).
Point 12: Table 2: the species must not be capitalized. What is ADR? Last column: the references go with first author et al., also in Table 3
Response 12: The modification was made as recommended in Table 2 and Table 3. Adverse drug reactions were spelled out.
Point 13: Line 113: why capital letters for gastrointestinal diseases?
Response 13: The modification (i.e., changing capital letters to lowercase letters) was made as recommended in the titles of 3.1, 3.2, and 3.3
Point 14: Line 126: species plural is abbreviated spp. not sp., for Yersinia it can be only Yersinia enterocolitica I think
Response 14: The modification was made as recommended (in line 142 in the revised manuscript).
Point 15: Line 128: reference number
Response 15: The reference number was added in new line 146.
Point 16: Line 151: might. Please note that when you refer to previous results or activities of researchers the correct form of verbs is the past tense.
Response 16: The modification was made as recommended (in line 167 in the revised manuscript).
Point 17: Line 167: delete "microbiota". It must be intestinal dysbiosis
Response 17: The modification was made as recommended (in line 184 in the revised manuscript).
Point 18: Line 192: of the E. faecalis strain of human origin...
Response 18: The modification was made as recommended (in line 210 in the revised manuscript).
Point 19: Line 222: "the production of bacteriocins by probiotic enterococcal consumption", it is not the consumption that produces bacteriocins, Please, adjust
Response 19: The modification was made as recommended (in lines 239-240 in the revised manuscript).
Point 20: Line 226: what is E. faecium liquid? E. faecium suspension, maybe? What is ARI?
Response 20: The modification was made as recommended (in line 243 in the revised manuscript). “Acute respiratory infections” was spelled out (in line 243-244 in the revised manuscript).
Point 21: Lines 230-231: italicize Salmonella and E. faecium
Response 21: The modification was made as recommended (in lines 247-248 in the revised manuscript).
Point 22: Line 248: GABA production
Response 22: The modification was made as recommended (in line 265 in the revised manuscript).
Point 23: Line 261: of enterococci, otherwise capitalize and italicize
Response 23: The modification was made as recommended (in line 278 in the revised manuscript).
Point 24: Lines 363-366: please, add a sentence to explain that these tetracycline resistance genes also confer resistance to tigcycline
Response 24: The modification was made as recommended (in lines 381-382 in the revised manuscript). The reference was updated as well.
Point 25: Line 371: antibiotic resistance genes
Response 25: The modification was made as recommended (in line 388 in the revised manuscript).
Point 26: Line 421: their beneficial effects on cholesterol reduction.
Response 26: The modification was made as recommended (in line 438 in the revised manuscript).
Point 27: Line 426: delete "beneficial species", you are referring to strains, not species
Response 27: The modification was made as recommended (in lines 443-444 in the revised manuscript).
Point 28: Lines 430-431: driven by the acquisition of virulence and antibiotic resistance genes.
Response 28: The modification was made as recommended (in line 448 in the revised manuscript).
I hope this addresses all your suggestions and comments. Please let us know if you have any further questions.

Round 3
Reviewer 2 Report
Some more corrections are needed as listed below.
The title sounds better as follows: Evaluation of Enterococcal Probiotic Usage and Review of Potential Health Benefits, Safety, and Risk of Antibiotic Resistant Strain emergence. No need for the first part of the current title
Lines 14-15: Please replace “but may increase the chance of multi-resistant isolates” with “but a thorough evaluation of virulence absence and risk of antibiotic resistance spread is needed at the strain level”.
Lines 33-34: found predominantly in the gut of humans and animals.
Line 73: to all L. acidophilus or B. subtilis strains.
Line 77: as probiotics
Line 78: , in probiotic products, efficacy,…
Line 103: that contain
Line 118: indicated
Line 124: This section discusses the probiotic potential of enterococci. "consumption" is too close to that at the end of the paragraph.
Line 352: please, delete "Despite concerns about these antibiotic resistances,"
Lines 420 and 429: in the transfer of genetic material the "recipient" is the strain that receives the genetic material. In these sentences humans are the host in which transfer can occur, not the recipients
Line 444:, please, delete "prospective"
Please, modify the last sentence as follows: studies commented in this review indicated that usage of enterococci as probiotics still requires extensive evaluation of the safety status of the strains used.
Usage of articles and verb tense should be improved
Author Response
Thank you, again, for your valuable time, commitment, and additional review of our manuscript. As always, we deeply appreciate your constructive feedback.
We have accepted all the detailed suggestions and modified the manuscript accordingly.
We will also attach a file that shows all the “track changes” on (a pdf file) in addition to the clean file for the third revision.
I hope this addresses all your suggestions and comments. Please let us know if you have any further questions.
